# Epidemiology of Sandfly-Borne Phleboviruses in North Africa: An Overview

**DOI:** 10.3390/insects15110846

**Published:** 2024-10-29

**Authors:** Sabrina Sellali, Ismail Lafri, Rafik Garni, Hemza Manseur, Mohamed Besbaci, Mohamed Lafri, Idir Bitam

**Affiliations:** 1Institute of Veterinary Sciences, University Blida 1, Blida 09000, Algeria; s_sellali@univ-blida.dz (S.S.); hemzamensouri@yahoo.fr (H.M.); besbaci@univ-blida.dz (M.B.); medlaffri@yahoo.fr (M.L.); 2Laboratory of Biotechnologies Linked to Animal Reproduction, Institute of Veterinary Sciences, University Blida 1, Blida 09000, Algeria; 3Department of Microbiology and Veterinary Pathology, Institut Pasteur of Algeria, Algiers 16000, Algeria; lafrismail@gmail.com; 4Department of Preventive Medicine, Institut Pasteur of Algeria, Algiers 16000, Algeria; ax19th@gmail.com; 5Center of Research in Agro-Pastoralism, Djelfa 17000, Algeria

**Keywords:** phlebovirus, Toscana virus, Sicilian virus, meningoencephalitis, *Phlebotomus* fever, Morocco, Algeria, Tunisia, Libya, Egypt

## Abstract

Phleboviruses transmitted by sandflies had been described in North Africa since seven decades ago. However, knowledge gaps still exist in the region. Therefore, a comprehensive literature search is performed to provide an overview of their evidence and distribution. At least eight phleboviruses transmitted by sandflies are present in North African region, namely Toscana, Sicilian, Naples, Cyprus, Punique, Utique, Saddaguia, and Medjerda Valley viruses. They were detected in sandflies, in animals, or in humans from Morocco, Algeria, Tunisia, Libya, and Egypt. Their infection rate ranged from 0.02% to 0.6% in sandflies and from 4.3% to 43.5% in dogs. Their overall seroprevalence in healthy persons was 15.9%. They were concentrated mainly in the north and might occur together. Sicilian (1.3–21%) and Toscana (3.8–50%) viruses were the most prevalent among humans. In particular, Toscana virus was frequent and involved in clinical cases from Algeria (3.8%) and Tunisia (12.86%). Noticeably, data from Morocco focused on TOSV in sandflies, whereas data from Libya and Egypt were very limited and old, respectively. Phleboviruses transmitted by sandflies are widespread and constantly present in North Africa. This calls for the implementation of phlebovirus diagnosis in patients with febrile and neurological manifestations.

## 1. Introduction

The last few decades have been characterized by the global emergence of many new infectious diseases in the human population caused by arthropod-borne viruses (arboviruses) [1,2]. Viral diseases are spreading quickly to new territories because of commercial transportation, globalization, deforestation, and urbanization. Climate and environmental changes induced by those disruptive human activities facilitate the spread of vectors and cause them to get closer to humans [3,4].

Phlebotomine sandflies (Diptera: *Psychodidae*) are regarded as important arthropods in human and veterinary medicine, as they harbor pathogens that infect humans and both domestic and wild animals [5,6]. They have a global distribution extending over southern Europe, Asia, Africa, Australia, and central and South America, with peak activity in the warmer months of the year [7]. Despite having a relatively limited geographic dispersal capacity compared with mosquitoes, sandflies exhibit an expansion of activity zones owing to changes in the climate and ecosystem, as exemplified by vector abundance and leishmaniasis cases [7,8]. Humidity is one of the key factors, together with suitable temperature, that define their geographical distribution in vast regions of both the old and new worlds [9]. They are often abundant in peri-urban, rural, and natural environments, which are concentrated near human and domestic animal habitats in suitable areas [10,11]. Therefore, further introduction of pathogens into naïve regions, leading to disease emergence and outbreaks, should be expected [6,7].

Although primarily known as proven vectors of protozoan parasite *Leishmania (Le.)*, sandflies also transmit other pathogens during their blood meal, namely bacteria of the genus *Bartonella* and viruses belonging to five different families: *Rhabdoviridae*, *Flaviviridae*, *Reoviridae*, *Peribunyaviridae,* and *Phenuiviridae* [5,6,12]. Viral pathogens vectored by sandflies mainly belong to the *Phlebovirus* genus within the *Phenuiviridae* family (order *Bunyavirales*), which is by far the most studied [5,13,14,15,16,17]. As provided by the International Committee on Taxonomy of Viruses (ICTV), these viruses are currently classified into species based on the threshold of 95% identity in the amino acid sequence of the ribonucleic acid (RNA)-dependent RNA polymerase (RdRp) protein [13]. Accordingly, *Phlebovirus* comprises about 70 named viruses, of which Toscana virus (TOSV), Sandfly Fever Naples virus (SFNV), Sandfly Fever Sicilian virus (SFSV), and Cyprus virus (CYPV) are most frequently detected in sandflies [13,18,19,20]. They have been documented as human pathogens, with symptoms spanning from fever to central nervous system involvement [21,22]. SFSV and SFNV are responsible for a self-resolving febrile illness called sandfly or *Phlebotomus* fever [23,24]. CYPV, a variant of the SFSV, causes an acute febrile disease [25,26]. TOSV displays a neurotropism distinct from other phleboviruses. It is unique as an agent of central nervous system infections that manifest as aseptic meningitis, encephalitis, and meningoencephalitis following the febrile period [21,27].

Sandfly-borne phleboviruses’ distribution and circulation peaks are closely related to vector presence and activity [28]. The Mediterranean countries have been identified as endemic regions for many sandfly-borne phleboviruses [27,29]. They are not only confined to southern Europe and the Middle East: there is significant evidence for their presence in North African countries [22,30,31,32,33,34,35].

The number of newly described sandfly-borne phleboviruses has been steadily growing in recent years [36]. Recent studies conducted in Mediterranean countries showed an increase in phlebovirus circulation with newly discovered viruses [36,37]. For instance, seven phleboviruses representing three distinct serocomplexes were screened in the endemic region of the eastern Mediterranean basin, with various phleboviruses in circulation together [22]. Unfortunately, no up-to-date review is available to summarize the current published data.

Although sandfly-borne phleboviruses had been described in North Africa from seven decades ago, knowledge gaps still exist in the region [38]. Indeed, phleboviruses remain obviously less studied in Africa in comparison to European and Asian countries [15,23,33,34,35,39,40,41]. Despite several reports of acute infections and preliminary findings of seroreactivity in humans and non-human vertebrates, information on phlebovirus exposure remains limited, and their health relevance is largely uncharacterized [22,36,42]. This unsatisfactory state of knowledge could also be due to fragmentary and rather poorly understood data when compared with vast knowledge of well-investigated regions.

As these diseases can easily spread to even uninfected areas, permanent monitoring is needed to control phleboviruses [42]. Understanding epidemiology contributes to public health protection by predicting virus transmission patterns and so preventing infection emergence. Herein, a comprehensive literature search is performed with the aim of providing an overview of the evidence and the distribution in hosts and geographic locations of sandfly-borne phleboviruses across North African region. Relevant online literature was sought on the academic search engines of Google Scholar and PubMed, using the terms sandfly-borne phlebovirus, Morocco, Algeria, Tunisia, Libya and Egypt. Gathered data were mapped with Esri Arcgis pro, Version 3.3, license acquired by Institut Pasteur of Algeria.

## 2. Sandfly-Borne Phleboviruses in Algeria

### 2.1. Sicilian Virus

Sandfly-borne phleboviruses were first described in Algeria two decades ago, as attention had been drawn to TOSV in the world. The first sandfly-borne phlebovirus investigated in Algeria was SFSV, after it had been shown to be circulating in Asia, Africa, and southern Europe. In 2006, a viral sequence closely related to that of SFSV was detected in a *Phlebotomus (Ph.) ariasi* from Larbaa Nath Iraten in the governorate of Tizi Ouzou (Kabylia), northern Algeria. Hence, sera from healthy persons living in the same locality were tested for SFSV immunoglobulin (Ig) G, of which 5% were positive [43].

### 2.2. Cyprus Virus

The sequence determined from *Ph. papatasi* trapped in northern Algeria in 2007 was most closely related to CYPV but not to the SFSV-like phlebovirus obtained from *Ph. ariasi* in the same region in 2006. No virological nor serological data were available to support this sporadic detection of Cyprus-like virus in Algeria [44].

### 2.3. Naples Virus

In 2007, a combination of entomological and seroprevalence studies provided evidence that a novel phlebovirus distinct from but related to SFNV was present in two areas, Larbaa Nath Iraten and Bou Ismail in the governorates of Tizi Ouzou and Tipaza, respectively, in northern Algeria, and could infect humans. Two sequences obtained from *Ph. longicuspis* were found to be most closely related to the Poona strain of SFNV and clearly belonged to the species SFNV. Human infection due to this virus was supported by seroprevalence studies indicating rates varying from 10.6 to 21.6%. The indirect immunofluorescence-based seroprevalence studies were carried out on healthy persons and patients attending biological laboratories for any reason in the vicinity of the areas where sandflies were trapped, respectively [44].

### 2.4. Toscana Virus

One strain of TOSV, named TOSV–Algeria 189, was isolated from sandflies collected in 2013, in Draa El Mizan, governorate of Tizi Ouzou. Phylogenetic studies indicated that it was most closely related with the TOSV strain from Tunisia within lineage A, which also includes Italian, French, and Turkish strains. A seroprevalence study performed on blood donor sera originating from the same area showed that almost 50% possessed TOSV-neutralizing antibodies, a much higher rate than that observed in southern Europe. Together, these results strongly suggested that TOSV was heavily affecting sandfly-exposed people in northern Algeria [31].

A further study has reported the results of the diagnosis of neuro-invasive TOSV infection in Algeria between 2016 and 2018 and described the first isolation of the TOSV strain from human samples in North Africa. TOSV infection was confirmed in 3.8% of hospitalized patients displaying neurological infection, mainly restricted to northern Algeria. TOSV cases were identified in coastal or inland governorates known for high activity of sandfly vectors and where TOSV-specific antibodies were detected in dogs, but not in the region of Kabylia, although it had previously been reported to be highly prevalent in humans there. The isolated strain was recovered from the cerebrospinal fluid (CSF) of a patient living in the governorate of Tlemcen, northwest of Algeria, and was genetically close to the strain isolated from sandflies collected in the region of Kabylia in 2013. These data confirmed that TOSV was endemic in Algeria and represented an important etiologic agent of neuro-invasive disease [34].

Regarding animals, studies were rather interested in dogs, as they were shown to be good markers for virus exposure and proxies for TOSV circulation [45]. In 2014, 4.3% of watchdogs from Tizi Ouzou and Bejaia governorates, northern Algeria, appeared to have neutralizing antibodies against TOSV [46]. A more recent study reported an Enzyme-Linked Immunosorbent Assay (ELISA)-based seroprevalence rate of 20% (14.5–25.5%) and suggested likely evidence for dog transmission and circulation of phleboviruses other than TOSV. Alongside this, a seroprevalence rate of 4.56% (1.65–7.43%) was recorded by a microneutralization test elucidating the exact occurrence of TOSV exposure in dogs. Eventually, these findings demonstrated that TOSV and antigenically related viruses were not only confined to Kabylia, or to northern Algeria, as believed at the outset of this study, but that they were even present in the south [45].

The latest findings provided an update of TOSV geographic distribution and revealed the exposure of different domestic animals other than canines, with a potential role of livestock (*p* = 0.00731) in the natural cycle. Positive results were obtained in 14.6% of cattle, 17.18% of sheep, 15% of horses, and 3.33% of goats. They originated mainly from the northern center of Algeria, with new areas where the virus was detected for the first time, namely Medea and Ain Defla in the north and Tissemsilt in the northwest [35].

### 2.5. Punique Virus

The impact of Punique virus (PUNV) on public health is still unknown. However, as a member of the SFNV group, of which the majority are pathogenic to humans, it may involve a potential risk for public health. An entomological study conducted in Kherrata, governorate of Bejaia, in 2020 resulted in the detection of two phleboviruses closely related to PUNV isolated in Tunisia and detected in Algeria. *Ph. perniciosus* (98.67%) was the dominant species and was identified in both positive pools. The overall prevalence of phlebovirus infection among the collected sandflies was 0.06%. The isolation for the first time of two different strains related to PUNV and the second report in Algeria from two distinct regions confirmed its large circulation in the country [47].

## 3. In Morocco

### 3.1. Toscana Virus

TOSV RNA was detected for the first time in Morocco in 2008, in *Ph. perniciosus* collected from Louata, a locality in Sefrou province, northern Morocco. The rate of infection in the vector *Ph. perniciosus* was (4/129) higher than rates reported in northern Mediterranean countries. The detected TOSV belonged to the genotype B, which has been previously recognized in the neighboring countries of France, Spain, and Portugal [48].

Subsequently, in the district of El Hanchane near Essaouira, center of Morocco, *Ph. sergenti*, the most abundant sandfly species (76%), was found to be infected by TOSV. The phylogenetic analysis indicated that the obtained sequence was most closely related to the TOSV France strain. So far, that is the first time that TOSV has been detected in phlebotomine sandflies other than *Ph. perniciosus*, *Ph. perfiliewi*, and *Se. minuta*. Hence, *Ph. sergenti* was considered to be a potential vector of TOSV [49].

In that same context, TOSV was obtained from *Ph. longicuspis* in Sefrou province and *Ph. sergenti* in Azilal province. Phylogenetic analyses indicated that those viruses clustered with TOSV strains circulating in Morocco, Spain, and France, i.e., belonged to genotype B [30].

Correspondingly, a much larger study that allowed exclusive identification of TOSV showed positive results in four provinces, Azilal, Sefrou, Essaouira, and Ouarzazate together, proving its existence in the north, the center, and the south of the country. In addition, TOSV was detected and isolated from three new sandfly vectors: *Ph. sergenti*, *Ph. longicuspis*, and *Ph. papatasi*. Three viruses were isolated from *Ph. sergenti* in Azilal, whereas two were found in *Ph. longicuspis* in Sefrou. The overall prevalence of TOSV infection in trapped sandflies was 0.21%. Sequence analysis confirmed that the Moroccan TOSV belonged to genotype B and was close to those isolated in France and Spain [50].

A more recent study reported evidence for TOSV in central Morocco. TOSV was detected in Lalla Laaziza locality from Chichaoua Province, where *Ph. sergenti* was the most abundant sandfly species (47.6%) [11].

### 3.2. Naples Virus

According to the unique study involving SFNV, neutralizing antibodies were found to occur in 2.9% people from Itzer in the province of Midelt, central Morocco [38].

### 3.3. Sicilian Virus

Scarce findings claimed that 5.7% people living in Itzer, province of Midelt, possessed SFSV-neutralizing antibodies (Nt-Ab) together with SFNV’s [38]. Shortly after, wild rodents and insectivora trapped in the north displayed a seroprevalence rate of 9.4% [51].

## 4. In Tunisia

### 4.1. Sicilian and Sicilian-like Viruses

The earliest study on this topic presented data on a selected human population in 1975, in which the SFSV-Nt-Ab rate was 1.6% [38]. Then, sera of wild rodents, insectivores, and cheiroptera, trapped in 1980 in different areas, mostly in Eastern Tunisia, were positive for antibodies against SFSV at a rate of 31% [52].

Three decades later, viral sequences corresponding to a novel phlebovirus closely related to but distinct from SFSV were detected in the same vector as PUNV within the region of Utique, northern Tunisia. The infection rate of sandflies was 0.53% for SFS-like viruses. Obtained sequences were related to SFSV, Corfu virus, SFSV Algeria, and SFSV Kabylia and were most closely related to Corfu virus. They were detected mainly from sandfly species belonging to the subgenus *Larroussius* (*Ph. perniciosus* and *Ph. longicuspis*), besides a sandfly species of the subgenus *Sergentomyia (Se. minuta)* [53].

Afterwards, the presence of SFSV was revealed in the CSF sample of one febrile patient with meningoencephalitis from the Sousse region, situated in the lower semi-arid bioclimatic zone. It showed similarities with SFSV isolated in Algeria. SFSV have infrequently been implicated in central nervous system infections. However, the presence of symptoms and the detection of viral RNA confirmed that the Sousse patient suffered from an acute infection with SFSV. A sporadic unexplained case of meningitis recorded during summer was then attributed to SFSV. Likewise, sporadic cases from Turkey caused by this virus or a SFS-like virus were described [54].

A next study provided evidence that detected phleboviruses in sandflies collected from central Tunisia in 2014 belonged to different clusters corresponding to SFSV and Utique virus (UTIV), as well as Saddaguia virus and TOSV. The development of irrigation in arid bio-geographical areas of central Tunisia may have led to an abundance of *Ph. perfiliewi* and thereby the emergence of phleboviruses in that region [25].

Moreover, a high rate of SFSV-NT-Ab (38.1%) was observed in dogs from the governorate of Kairouan (central Tunisia), demonstrating that SFSV could infect dogs and circulated at high levels in the region [32].

Again in central Tunisia, a seropositivity of 1.3% was recorded for SFSV among recently assayed blood donors from Kairouan and thereabouts. However, exposure to sandfly-borne phleboviruses other than TOSV was not identified in the Kairouan governorate but was in Sousse and Monastir [55].

### 4.2. Cyprus Virus

The first and the only seroepidemiology survey testing for human exposure to CYPV concerned the general population of central Tunisia during 2017 and disclosed a seroprevalence rate of 2.9% [55].

### 4.3. Punique Virus

A molecular analysis carried out in 2008 revealed the presence of a novel virus closely related to SFNV which was named Punique virus as reference to its origin, a Punique ruin site. PUNV was isolated from *Ph. perniciosus* and detected in *Ph. longicuspis* from the Utique region. The infection rate of sandflies with SFN-like viruses was 0.13% [53].

Subsequently, sandflies were collected during two other consecutive years from the site in Utique. In 2009, PUNV was detected in *Ph. perniciosus*, yielding an infection rate of 0.11%, while in 2010, it was found to be carried in unidentified sandflies with an infection rate of 0.05%. The isolated strains were most closely related with the sequence of the PUNV strain isolated in 2008 [33]. The new PUNV strains were obtained together with TOSV that was likely cocirculating [56]. The phenology of sandflies in the site of Utique showed that *Ph. perniciosus* was the most abundant sandfly species and suggested that it was the main vector of PUNV. PUNV was isolated from sandflies of northern Tunisia during three consecutive years, 2008, 2009, and 2010, supporting the idea that it is endemic in this geographic area [33].

To attempt the determination of the role of PUNV in human infection, a sero-epidemiological study among a population living in the governorate of Bizerte, northern Tunisia, was performed in the same period. PUNV-NT-Ab were detected in 8.72% of tested sera. Accordingly, PUNV could infect humans, but it occurred seldom in a region where the virus circulated at a high level in sandfly populations. Furthermore, it was involved at a much lower rate in human infections in northern Tunisia compared to TOSV [57]. Similarly, PUNV was isolated from *Ph. perniciosus* collected in Medjez el Bab, located in the governorate of Beja, between 2011 and 2012 [54].

Further evidence that PUNV was present in Tunisia was provided by a neutralization-based seroprevalence in dogs from the governorate of Kairouan, central Tunisia. The results showed the cocirculation in the region of TOSV and SFSV with PUNV, of which the seroprevalence rate was 43.5%. In particular, PUNV much more frequently infected dogs than other phleboviruses [32].

### 4.4. Naples Virus

According to a recent seroprevalence study focusing on the Tunisian central governorates, exposure to SFNV in blood donors was estimated as 1.1%. Therein, SFNV seroprevalence was the lowest among the five phleboviruses assayed for. Unfortunately, there have been no studies so far that could be considered as a further elaboration [55].

### 4.5. Toscana Virus

The first report of TOSV circulating in Tunisia aimed to evaluate its implication in neurological disease patients between 2003 and 2009. Specific IgM for TOSV were detected in 10% of patients with neurological diseases that predominated during summer and autumn and originated, for the majority, from the coastal region. Anti-TOSV IgG were detected in 7% of cases, corresponding to previous infection [58]. TOSV was even observed to circulate regularly throughout the study period, which suggested that its circulation was continuous [59].

Further, TOSV-NT-Ab were detected in 41% of out-care patients from the governorate of Bizerte. More importantly, TOSV appeared responsible for the vast majority of human infections by sandfly-borne phleboviruses in northern Tunisia [57].

Later, patients with meningeal syndrome from the four cardinal points of Tunisia were investigated. More than 12% patients were IgM-positive for TOSV, among whom 78% were IgG-positive. Yet 12.86% of CSF samples contained TOSV RNA and were mostly from Tunis. Along with this, *Ph. perniciosus* and *Ph. perfiliewi* trapped in northern Tunisia near the habitats of some TOSV-positive patients showed evidence for TOSV. The obtained sequences corresponded to those described in Morocco, France, Italy, and Tunisia. Several unexplained cases of meningitis recorded every summer in Tunisian hospitals were then confirmed as being due to TOSV [54].

In 2010, sandflies from Utique were found to be positive for TOSV (and PUNV), yielding an infection rate of 0.03%. Most of the sandflies belonged to the subgenus *Larroussius* (98.3%), and *Ph. perniciosus* was the most abundant species (71.74%). It is therefore probable that TOSV was transmitted by sandfly species of the subgenus *Larroussius*. The TOSV strain obtained from positive pools was most closely related to strains within the Italian lineage [56].

As for animals, a retrospective study of TOSV and other phleboviruses, namely PUNV and SFSV, was undertaken in dogs from the governorate of Kairouan during 2013. Only 7.5% of dog sera possessed TOSV-NT-Ab. Thus, TOSV was present in dogs from the governorate of Kairouan but at a lower rate compared to PUNV and SFSV. Indeed, this was in contrast with the high rate of TOSV-NT-Ab and low rate of PUNV-NT-Ab reported in humans from the governorate of Bizerte [32].

The first detection of TOSV from sandflies collected from central Tunisia was reported in 2014. It suggested that the abundance of *Ph. perfiliewi* was associated with the development of irrigation in arid bio-geographical areas of central Tunisia, which may have led to the emergence of phleboviruses [25].

Entomological screening performed in the village of Saddaguia, a highly irrigated area located in the governorate of Sidi Bouzid within arid central Tunisia, between 2014 and 2016 revealed the presence of TOSV, with an infection rate of sandflies varying from 0.05% to 0.22%. The three-year investigation clearly showed the endemic co-circulation of TOSV and *Le. infantum*. However, no co-infection of TOSV and *Le. infantum* was detected in any of the sandfly pools investigated. According to the phylogenetic tree topology, the Tunisian TOSV strains isolated in 2015 and those previously described in 2010 were grouped within one phylogenetic cluster (sublineage A) and were very close to each other [60].

On another note, newly reported updates showed that the percentage of IgG positivity for TOSV was 13.3% in asymptomatic blood donors from central Tunisia. It has been by far the most frequently detected virus among the six tested from the order *Bunyavirales* [55].

### 4.6. Saddaguia Virus

An entomological investigation performed in 2013 highlighted the presence of a novel phlebovirus in Tunisia, yielding a minimum infection rate of sandflies of 0.6%. This putative novel virus, tentatively called Saddaguia virus (SADV), was clearly distinct from but closely related to Massilia and Granada viruses. SADV was widely distributed in Tunisia. Viral sequences were detected in sandflies collected from Mateur and Borj Youssef in the north and Saddaguia in the center, where *Ph. perniciosus*, *Ph. perfiliewi*, and *Ph. longicuspis* were the most abundant species [24]. A further study performed in the same region showed the coexistence of SADV with TOSV, SFSV, and UTIV in sandflies [25].

### 4.7. Medjerda Valley Virus

Medjerda Valley virus (MVV) was isolated from *Phlebotomus* spp. sandflies trapped during the summer of 2010 in the vicinity of the Utique site (northern Tunisia), yielding an infection rate of 0.02%. Genetic analysis indicated that MVV was most closely related to members of the Salehabad virus species, being the first evidence of a virus within the Salehabad virus species in Tunisia. The novel phlebovirus was tentatively named MVV from the eponymous valley located near the trapping site. MVV-NT-Ab were detected in 1.35% of outpatient sera from the same region, which demonstrated that MVV can readily infect humans despite low seroprevalence rates [61].

## 5. In Libya

Unfortunately, sandfly-borne phleboviruses are very poorly documented in Libya. Antibodies for two types of sandfly viruses were detected in a serological survey conducted in 2008, indicating past exposure. It showed the presence of antibodies in humans against SFNV and SFSV but with very low prevalence rates (0.5% and 0.7%, respectively). Thus, it was simply meant to provide a baseline for further studies [62].

In 2014, local circulation of TOSV was revealed in 25% of patients attending hospital for laboratory investigations in the area of Yafran, located in Jabal Nafusah (northwestern Libya), an endemic area of leishmaniasis [63].

## 6. In Egypt

Sandfly-borne phlebovirus studies in Egypt were exclusively long-established and rudimentary. They had covered period between the 1950s and early 1990s, when only SFNV and SFSV were reported. They were repeatedly isolated from sandflies *(Ph. papatasi)* and acutely ill patients, and their seroprevalence in humans was 2–56.3% for SFNV and 2–59.4% for SFSV. Many years later, SFSV was presumptively considered a probable cause for fatal encephalitis, although there was insufficient evidence to support the virus clinical case of the IgM-positive patient [23,64].

## 7. Discussion

This study provides overall synthesized data on occurrence and recovery of sandfly-borne phleboviruses in North Africa, based on available research and conclusions made in original evidence papers, since 1971. The reviewed data revealed that sandfly-borne phleboviruses are widespread in North Africa, with eight virus species detected in either sandflies, humans, or animals between 1950 and 2020 in the five neighboring countries of Morocco, Algeria, Tunisia, Libya, Egypt together. Sandfly-borne phleboviruses were repeatedly, but unequally, studied in the North African countries. There was remarkably more emphasis on Tunisia, where more numerous current studies were realized, showing evidence for various phlebovirus species, compared to Algeria and Morocco (Table 1). In Libya, there is a scarcity in studies about the occurrence of sandfly-borne phleboviruses. The lack of case reports may be due to weak knowledge of the actual virus epidemiology in the country and shortage of laboratory diagnosis facilities [63]. No current data were available from Egypt. The collected data might be regarded as out of date and need to be updated for more relevance [23]. Thusly, few conclusions could be drawn on these two North African countries.

TOSV, followed by SFSV and PUNV, was subjected to more important findings from North Africa. They were likely most commonly investigated in the region because of their potential pathogenicity [12,15,65], their expansion [32,34,35,45,57], or their historical background [38,43,51,52].

According to the latest review, TOSV was confirmed to circulate all over the Mediterranean area. In addition to North African countries, TOSV has been reported in Portugal, Spain, France, Italy, Turkey, Croatia, and Greece, but also in Balkan countries such as Kosovo, Bosnia Herzegovina, and Bulgaria and in the Mediterranean islands of Elba, Baleares, Malta, Corsica, Sardinia, Cyprus, and Crete. Human seroprevalence of TOSV was usually around 10–24%; however, it exceptionally reached 77.2% in a high-risk population from Italy (Tuscany) and 37.5% in a resident population in Croatia who presented IgM antibodies. The first isolation of TOSV from animals was from a brain sample of a bat *(Pipistrellus kuhlii)* captured in 1984 in Tuscany, Italy. Then, it has been detected in other mammals and poultry from Portugal, France, Greece, Turkey, and Spain [23,41,66].

Human infections of Sicilian or Sicilian-like phleboviruses have been reported in many countries of the Mediterranean region and the Middle East. Outbreaks or sporadic human cases have been described, for example, in Cyprus, Turkey, Iraq, and Ethiopia. In Europe, cases have been reported from France, Italy, Greece, and Kosovo. Likewise, there were studies in Africa from Sudan and in Asia from Iran, Pakistan, Bangladesh, and Afghanistan. Regarding the circulation of Sicilian phlebovirus among non-human vertebrates, several reports confirmed its detection in different mammal species, including rodents, insectivores, and carnivores in Spain and Italy, bats in Spain, dogs in Greece and Cyprus, and livestock in Kosovo [41].

In Algeria, the phleboviruses detected in sandflies were TOSV, SFSV, CYPV, and PUNV [31,43,44,47], but infection rate was only available for the latter (0.06%), which was almost equal to the rate reported in Tunisia (0.05%) for the same virus [33,47]. Sandflies from Tunisia yielded variable rates of infection with TOSV and PUNV, ranging, respectively, from 0.03% to 0.22% [56,60] and from 0.05% to 0.13% [33,53]. The presence of other phlebovirus species in sandflies from Tunisia was described as well. MVV was less prevalent (0.02%); however, its infection rate remained consistent with the formers [61]. On the contrary, infection rates for SADV (0.6%) and UTIV (0.53%) were superior and appeared to be the highest among the phleboviruses circulating in sandflies in the whole region of North Africa [24,53]. In Morocco, less diversity was documented. TOSV alone was present in sandflies, with an infection rate (0.21%) similar to that reported in sandflies from Tunisia (0.22%) for the same virus [11,30,48,49,50,60].

It is worth underlining that *Ph. perfiliewi*, *Ph. perniciosus*, and *Ph. longicuspis* were the most common phlebovirus vectors in North Africa. They were often involved in the transmission of various phleboviruses such as SFSV, TOSV, PUNV, UTIV, and SADV. In particular, *Ph. perniciosus* and *Ph. longicuspis* were known to circulate and harbor phleboviruses in three North African countries (Morocco, Algeria, and Tunisia; Table 1). In fact, the main sandfly vector of TOSV is believed to be *Ph. perniciosus* [41,42]. It occurs mainly from the humid to arid bioclimatic zone but can also be found in the Saharan bioclimatic zones in the south area and central Sahara, albeit with low density compared to the northern regions. *Ph. longicuspis* is generally recorded in sympatry with *Ph. perniciosus* from the subhumid to Sahara bioclimatic zones. *Ph. perfiliewi* is found in subhumid, arid, and subarid regions at high latitudes. It was sympatric with *Ph. longicuspis* and *Ph. Perniciosus* in Morocco, Algeria, and Tunisia. These countries share the same climatic conditions and landscape environments in large borderland areas, allowing the exchange of species either naturally or via human activities [67]. Importantly, weather conditions, namely precipitation, were shown to have a direct influence on the abundance of sandflies [40].

Animal investigations conducted on sandfly-borne phleboviruses in North Africa concerned livestock and small wild mammals, but focused more importantly on dogs (Figure 1) due to their supposed role in phlebovirus transmission cycles [68]. Findings were exclusively serological and provided no evidence for virus replication, which impeded any contribution to reservoir host identification from the region. TOSV seroprevalence rates in dogs from Algeria (4.3%, 4.56%) and Tunisia (7.5%) were low and close, whereas those of SFSV (38.1%) and PUNV (43.5%), whose distribution was geographically restricted to Tunisia, owing to the lack of corresponding documentations in other countries, were significantly more prevalent [32,45,46]. Surprisingly, in Tunisia, SFSV seroprevalence observed recently in dogs (38.1%) was congruent with that obtained from small wild mammals (31%) analyzed four decades ago. According to this same study context applied in Morocco, SFSV seroprevalence (9.4%) was much lower [32,51,52].

Assessed human sera, either healthy or patients’, were mostly positive for anti-SFSV and anti-TOSV Ab (Figure 1 and Figure 2). Occurrence of antibodies against other sandfly-borne phleboviruses in studied persons was sporadic and lesser, including SFNV (2.9%) in Morocco and (1.1%) Tunisia and PUNV (8.72%), CYPV (2.9%), and MVV (1.35%) in Tunisia [38,55,57,61]. As claimed by an overall study in Tunisia, 15.9% of the healthy population as IgG-positive against sandfly-borne phleboviruses, of which 51% resided in rural areas. Dual reactivity was observed between SFSV and SFCV (0.8%) and between TOSV and SFNV (0.5%) or SFCV (0.5%) [55].

SFSV seroprevalence in North African population varied from 1.3% to 21%, while TOSV occurrence was between 3.8% and 50%, showing a huge gap between confirmed etiologic agent prevalence and asymptomatic infection rate, respectively (Table 1). Particularly, TOSV was identified as an etiological agent of neurological disease in patients from Algeria and Tunisia with notable rates (3.8% and 12.86%, respectively) [34,54]. In contrast, it has never been described in humans from Morocco up to now because no studies were run for that purpose. Comparatively, data on sandfly-borne phleboviruses were lesser in Morocco. Studies conducted on human and non-human vertebrates were very limited and old. Recent investigations were exclusively entomological and concerned TOSV only (Table 1).

The discrepant seroprevalence would be due to the vector distribution among the highly changing sampling areas, which is tightly related to geographical area and environmental factors [31,32,45]. Sandfly-borne phleboviruses occurred in the arid and humid bioclimatic stages but were restricted mainly to the north. It is strongly suggested that phleboviruses are heavily affecting highly populated regions, like the northern and the coastal areas, whose sandfly-exposed inhabitants are at greater risk of infection [31,34]. Their distribution is congruent with entomological data. Sandflies are abundant in peri-urban and rural environments and are often found in close proximity to human and domestic animal habitats, which results in an overlap [35,45].

However, not all persons in a community have equal exposure to sandflies; some people apparently escape infection. Sandflies often show a marked atmosphere preference. This may indicate that some individuals within a community are at greater risk of virus infection than others [38]. In the Maghreb region, a total of 32 species of the genera *Phlebotomus* and *Sergentomyia* were reported (15 from Libya, 18 from Tunisia, 23 from Morocco, and 24 from Algeria). They feed on a wide range of vertebrate hosts, and their trophic preferences vary depending on the sandfly species. Multiple feeding patterns were detected, with various combinations of humans and vertebrate hosts [64]. As example, *Ph. perniciosus* is more anthropophilic compared to *Ph. longicuspis*, which is more zoophilic, leading to a difference in human versus dog biting rates and subsequently to a difference in the infection status with phleboviruses [32].

More significantly, sandfly fauna density and diversity vary by bioclimatic zones [54]. The number of sandfly vectors, namely *Ph. sergenti*, *Ph. longicuspis*, and *Ph. perniciosus*, showed a positive correlation with altitude. In fact, the biodiversity index and richness were higher in highland areas (912 m above sea level) than in plain areas. Along with the abiotic factor of altitude, the distribution, biodiversity, and richness of sandflies seemed to be primarily influenced by bioclimatic factors such as the effect of soil texture on their habitats [11].

In addition, latest findings showed that environmental changes, mainly due to the intensive development of irrigation systems for agriculture implemented in arid bio-geographical areas, yielded sustained populations of sandflies of the subgenus *Larroussius*, mainly *Ph. perfiliewi*, *Ph. perniciosus*, and *Ph. Longicuspis*, which may have led to the emergence of phleboviruses in new areas [25,60].

Recent data from elsewhere in the African continent were only available in Kenya, where a novel virus designated Ntepes virus (NPV) was isolated from sandflies and specific neutralizing antibodies were found in 13.9% of humans from the northeast [69].

Furthermore, in Italy, at least six identified phleboviruses co-circulated in sandflies: TOSV, Fermo (FERV), Corfou (CFUV), Ponticelli I (PONV-I), Ponticelli II (PONV-II), and Ponticelli III (PONV-III) [36,40,70]. On the other hand, co-circulation of phleboviruses and *Leishmania* parasites occurred in sandflies, of which *Ph. perfiliewi* (98%) was the foremost species [40]. Still, the results of a serological survey performed on domestic animals, against TOSV, FERV, PONV-I, and PONV-III, showed neutralizing activities for at least one of the four phlebovirus strains in 17.3% of dogs, 57.1% of goats, and 18.7% of sheep [71]. For TOSV alone, isolation and high seroprevalence rates (30–50%) were reported in a population from several regions of the country [23].

Four phleboviruses are known to circulate in Portugal: Sicilian, Toscana, Alcube, and Massilia phlebovirus [41]. For TOSV, Portugal was the second country, after Italy, to be considered endemic, with seroprevalence rates of 4.2% and 1.3% in patients with and without neurological symptoms, respectively, and 5.6% of CSF samples from meningitis patients being positive for TOSV infection. The presence of TOSV in animals was corroborated by several seroprevalence studies, which found antibodies in domestic animals such as cats (2.2–4.9%) and dogs (6.2–56.3%), as well as in wild animals, namely wolves (*Canis lupus signatus*, 2%) and foxes (*Vulpes vulpes*, 2.7%). Sandfly vectors identified in the region were *Ph. perniciosus*, *Ph. papatasi*, *Ph. ariasi*, and *Se. minuta* [23,41].

In France, TOSV isolation was achieved from human samples and from *Ph. perniciosus* sandflies in Marseille and RNA detection from *Se. minuta*. Alongside this, 8.7–14% of blood donors possessed anti-TOSV IgG. For Sicilian virus, low seroprevalence rates were reported in the southwest (2%) and in Marseille (1%) among blood donors. Wild mammals displayed 0.3% seropositivity [23].

In Spain, TOSV seropositivity rates were lower (5–26%) than those reported in Italy. Seroprevalence studies conducted in domestic animals from Granada showed evidence that they were frequently bitten by infected sandflies (17.7% in goats, 17.9% in cows, 22% in pigs, 32.3% in sheep, 48.3% in dogs, 59.6% in cats, and 64.3% in horses) [23]. More recently, high neutralizing antibody rates were identified in wild animals, represented by the Common Quail *(Coturnix coturnix)* for SFSV (45.45%) and TOSV (42.45%) [66].

In Greece, serological rates were 13.1–24.7% for SFNV, 2–8.5% for SFSV, and 11–51.7% for TOSV. Peculiarly, neutralizing activity against SFNV and SFSV was noticed in farmers. Also, antibodies against Corfu virus/Sicilian virus were detected in 4% of healthy residents [23].

In Germany, competent vectors of TOSV were distributed in the southwest, and TOSV-neuroinvasive was detected in 4% of cases with suspected viral meningoencephalitis [72].

Far away, in the Middle East, the most frequently observed virus exposure in Turkey was due to TOSV (13.7–22.6%), followed by SFNV (5.2–15.3%), SFSV (12.1–14.7%), and Sandfly Fever Turkey Virus (12.1%) [22,23]. In particular, TOSV seroreactivity among blood donors had reached very high IgG (56%) and IgM (43.6%) antibody levels. Among central nervous system patients, TOSV IgM antibodies were detected in 11.2% of the sera and in 1.76% of the CSF samples, whereas IgG antibodies were detected in 8% of the sera and 3% of the CSF samples. Furthermore, IgM antibodies against SFSV (27.27–36%), SFNV (45.45%), and CYPV (4%) viruses were detected in acute patient sera [23]. Additionally, TOSV sequences were identified in birds’ organs during the screening of avian specimens collected in the Mediterranean coast of the Anatolian peninsula [41].

Nearby, sandfly fever outbreak was reported from northern Lebanon during 2007. SFNV appeared as the main infection and was detected in soldiers returning from Syria and Lebanon in 2019 [73,74].

Seroprevalence rates of 9.4% and 0.8% were found in blood donors from western Saudi Arabia for SFSV and TOSV, respectively. The highest seropositivity rate was among samples collected from animal handlers, suggesting that contact with livestock animals could be a risk factor. Sera from livestock animals showed seropositivity of 53.3% and 4.4% in cows, 27.5% and 7.8% in sheep, 2.2% and 0.0% in goats, and 10.0% and 2.3% in camels for SFSV and TOSV, respectively [75].

Epidemiological patterns observed with sandfly-borne phleboviruses imply that they have the same ecology. Their geographic distribution was quite similar and appeared to parallel that of the vectors occurring in humid zones (Figure 2), i.e., northern areas characterized by Mediterranean climate or highly irrigated arid areas [25,34,35,50].

Interestingly, either cutaneous or zoonotic visceral leishmaniasis, or both, were endemic in most of regions where sandfly-borne phleboviruses circulated, indicating an overlapping geographic distribution. It is important to highlight that most study areas consisted of leishmaniasis foci, which promoted evidence to *Leishmania* spp. and phlebovirus cocirculation, along with identification of common harboring sandfly species [11,32,33,38,44,46,50,53,56].

**Table 1 insects-15-00846-t001:** Summary of published studies of sandfly-borne phleboviruses in North Africa.

Country	Virus	Year	Area	Site Factors	Governorate	Region	Population	Prevalence	Ref.
Algeria	SFSV	2006	Larbaa Nath Iraten	NA	Tizi Ouzou	North	*Ph. ariasi*	NA	[43]
							Healthy humans	5% (IgG)	
	SFSV	2007	Bou Ismail	NA	Tipaza	North	*Ph. longicuspis*	NA	[44]
			Larbaa Nath Iraten	NA	Tizi Ouzou	North			
	SFSV	2007	Bou Ismail	NA	Tipaza	North	Outpatients	10.6% (IgG)	[44]
			Larbaa Nath Iraten	NA	Tizi Ouzou	North	Healthy persons	21.6% (IgG)	
	CYPV	2007	Bou Ismail	NA	Tipaza	North	*Ph. papatasi*	NA	[44]
			Larbaa Nath Iraten	NA	Tizi Ouzou	North			
	TOSV	2013	Draa El Mizan	36°32′146” N, 3°50′850” E 380 m altitude	Tizi Ouzou	North	Unidentified sandflies	NA	[31]
							Blood donors	50% (Nt-Ab)	
	TOSV	2014	Tifra	36°39”59” N, 4°22”16” E	Bejaia	North	Watchdogs	4.3% (Nt-Ab)	[46]
			Ouaguenoun	36°46”12” N, 4°10”29” E	Tizi Ouzou	North			
			Azazga	36°44”43” N, 4°22”16” E	Tizi Ouzou	North			
	TOSV *	2017–2018	NA	NA	Tlemcen	North	Owned dogs	20% (IgG)	[45]
			NA	NA	Blida	North			
			NA	NA	Algiers	North			
			NA	NA	Medea	North			
			NA	NA	Bouira	North			
			NA	NA	Setif	North			
			NA	NA	Bejaia	North			
			NA	NA	Laghouat	Center			
			NA	NA	Tamanrasset	South			
	TOSV	2017–2018	NA	NA	Blida	North	Owned dogs	4.56% (Nt-Ab)	[45]
			NA	NA	Algiers	North			
			NA	NA	Bejaia	North			
			NA	NA	Setif	North			
			NA	NA	Laghouat	Center			
	TOSV	2016–2018	NA	NA	Tlemcen	North	Hospitalized patients with neurological infection	3.8% (RNA/Ig)	[34]
			NA	NA	Blida	North			
			NA	NA	Medea	North			
			NA	NA	Algiers	North			
			NA	NA	Jijel	North			
			NA	NA	Setif	North			
			NA	NA	Batna	North			
			NA	NA	Biskra	North			
			NA	NA	Guelma	North			
			NA	NA	Oum El Bouaghi	North			
			NA	NA	Tebessa	North			
	TOSV	2017	NA	NA	Tissemsilt	North	Livestock	3.33–17.18% (Nt-Ab)	[35]
			NA	NA	Ain Defla	North			
			NA	NA	Tipaza	North			
			NA	NA	Blida	North			
			NA	NA	Medea	North			
			NA	NA	Algiers	North			
	PUNV	2020	Kherrata	36°24′20′′ N, 5°16′37′′ E	Bejaia	North	*Ph. perniciosus*	0.06%	[47]
Morocco	SFNV	1976	Itzer	NA	Midelt	Center	Humans	2.9% (Nt-Ab)	[38]
	SFSV	1976	Itzer	NA	Midelt	Center	Humans	5.7% (Nt-Ab)	[38]
	SFSV	1979	Bab Besen	1600 m altitude, Humid	Chaouen	North	Rodents Insectivora	9.4% (IgG)	[51]
			Chaouen	600 m altitude, Sub-humid	Chaouen	North			
			Talembote	400 m altitude, Semi-arid	Chaouen	North			
			Arbaoua	130 m altitude, Sub-humid	Kenitra	North			
			Asilah	200 m altitude, Sub-humid	Tanger	North			
	TOSV	2008	Louata	338,319 latitude, 4828 longitude, 800 m altitude	Sefrou	North	*Ph. perniciosus*	4/129	[48]
	TOSV	2011	El Hanchane	31°31′11′′ N, 9°26′02′′ W	Essaouira	Center	*Ph. sergenti*	NA	[49]
	TOSV	2008–2011	NA	NA	Azilal	Center	*Ph. sergenti*	NA	[30]
			NA	NA	Sefrou	North	*Ph. longicuspis*	NA	
	TOSV	NA	NA	Bani Hassan: 31°59′50′′ N, 6°59′44′′ W 722 m altitude, Semi-arid	Azilal	Center	*Ph. longicuspis*	0.18%	[50]
				Tabia: 32°01′49′′ N 6°47′48′′ W 563 m altitude Semi-arid			*Ph. sergenti*		
			El Hanchane	290–300 m altitude Semi-arid	Essaouira	Center	*Ph. sergenti*	0.15%	
			Ouarzazate	1134 m altitude Arid	Ouarzazate	South	*Ph. papatasi*	0.09%	
			Louata	674 m altitude	Sefrou	North	*Ph. longicuspis*	0.33%	
				Temperate			*Ph. perniciosus*		
	TOSV	2018	Lalla Laaziza	31°04′ N, 8°42′ W >912 m altitude	Chichaoua	Center	*Ph. sergenti*	NA	[11]
Tunisia	SFSV	1975	NA	NA	Mixed	NA	Humans	1.6% (Nt-Ab)	[38]
	SFSV	1980	El Gharia	750 m altitude	Bizerte	North	Rodents Insectivora Cheiroptera	31% (IgG)	[52]
			Zarzis	1 m altitude	Medenine	South			
			Kerkennah	1–3 m altitude	Sfax	Center			
			Tataouine	190 m altitude	Tataouine	South			
			Ezzriba	70 m altitude	Zaghouan	North			
			Moghrane	110 m altitude	Zaghouan	North			
	SFSV	2011–2012	Sousse	NA	Sousse	North	Patient with meningo- encephalitis	NA	[54]
	SFSV	2013	Haffouz	34°51′ N, 9°29′ E Arid	Kairouan	Center	Dogs	38.1% (Nt-Ab)	[32]
			Bouhajla	35°24′ N, 9°56′ E Arid					
	SFSV	2014	El Felta	35°16′ N, 9°26′ E	Sidi Bouzid	Center	*Ph. perfiliewi*	NA	[25]
	SFSV	2017	NA	NA	Monastir	Center	Blood donors	1.3% (IgG)	[55]
			NA	NA	Sousse				
	PUNV	2008	Utique	37°2′ N, 10°2′ E Sub-humid	Bizerte	North	*Ph. perniciosus*	0.13%	[53]
							*Ph. longicuspis*		
	PUNV	2009	Utique	37°08′ N, 7°74′ E	Bizerte	North	*Ph. perniciosus*	0.11%	[33]
	PUNV	2010	Utique	37°08′ N, 7°74′ E	Bizerte	North	Unidentified sandflies	0.05%	[33]
	PUNV	2011	NA	NA	Bizerte	North	Humans	8.72% (Nt-Ab)	[57]
	PUNV	2011–2012	Medjez El Bab	Semi-arid	Beja	North	*Ph. perniciosus*	NA	[54]
	PUNV	2013	Haffouz	34°51′ N, 9°29′ E Arid	Kairouan	Center	Dogs	43.5% (Nt-Ab)	[32]
			Bouhajla	35°24′ N, 9°56′ E Arid					
	UTIV	2008	Utique	37°2′ N, 10°2′ E Sub-humid	Bizerte	North	*Ph. perniciosus*	0.53%	[53]
							*Ph. longicuspis*		
							*Se. minuta*		
	UTIV	2014	Saddaguia	35°05′ N, 9°25′ E	Sidi Bouzid	Center	*Ph. perfiliewi*	NA	[25]
	SFNV	2017	NA	NA	Monastir	Center	Blood donors	1.1% (IgG)	[55]
			NA	NA	Sousse				
	TOSV	2003–2009	NA	NA	Mahdia	North	Neurological disease patients	10% (IgM) 7% (IgG)	[58]
			NA	NA	Monastir	North			
			NA	NA	Sousse	North			
			NA	NA	Sfax	Center			
			NA	NA	Gabes	South			
			NA	NA	Djerba	South			
	TOSV	2010	Utique	37°08′ N, 7°74′ E	Bizerte	North	*Ph. perniciosus*	0.03%	[56]
	TOSV	2011	NA	NA	Bizerte	North	Outpatients	41% (Nt-Ab)	[57]
	TOSV	2011–2012	NA	NA	Tunis	North	Patients with meningeal syndrome	12.16% (IgM) 78% (IgG) 12.86% (RNA)	[54]
							*Ph. perniciosus*	NA	
							*Ph. perfiliewi*	NA	
	TOSV	2013	Haffouz	34°51′ N, 9°29′ E Arid	Kairouan	Center	Dogs	7.5% (Nt-Ab)	[32]
			Bouhajla	35°24′ N, 9°56′ E Arid					
	TOSV	2014	Saddaguia	35°05′ N, 9°25′ E	Sidi Bouzid	Center	*Ph. perfiliewi*	0.03%	[25]
	TOSV	2014–2016	Saddaguia	35°05′ N, 9°25′ E	Sidi Bouzid	Center	*Ph. perniciosus*	0.05–0.22%	[60]
							*Ph. longicuspis*		
							*Ph. perfiliewi*		
	TOSV	2017	NA	NA	Kairouan	Center	Blood donors	13.3% (IgG)	[55]
			NA	NA	Monastir				
			NA	NA	Sousse				
	SADV	2013	Borj Youssef	Borj Youssef: 36°56′ N, 10°07′ E, Semi-arid	Ariana	North	*Ph. perniciosus*	0.6%	[24]
				Utique: 37°08′ N, 7°74′ E, Semi-arid			*Ph. longicuspis*		
	SADV	2013	Mateur	Mateur: 37°03′ N, 9°28′ E, Subhumid	Bizerte	North	*Ph. perfiliewi*		
				Sejnene: 36°56′ N, 9°21′ E, Humid			*Ph. perniciosus*		
	SADV	2013	Saddaguia	Saddaguia: 35°05′ N, 9°25′ E, Arid	Sidi Bouzid	Center	*Ph. perfiliewi*		
				Bouhajla: 35°24′ N, 9°56′ E, Arid			*Ph. perniciosus*		
	SADV	2014	El Felta	35°16′ N, 9°26′ E	Sidi Bouzid	Center	*Ph. perfiliewi*	NA	[25]
	MVV	2010	Utique	37°08′ N, 7°74′ E	Bizerte	North	*Phlebotomus* spp.	0.02%	[61]
							Outpatients	1.35% (Nt-Ab)	
	CYPV	2017	NA	NA	Monastir	Center	Blood donors	2.9% (IgG)	[55]
			NA	NA	Sousse				
Libya	SFNV	2008	NA	NA	NA	NA	Febrile patients	0.5% (IgG)	[62]
	SFSV	2008	NA	NA	NA	NA	Febrile patients	0.7% (IgG)	
	TOSV	2014	Yafran	32°08′ N, 12°55′ E 691 m altitude Semi-arid	Jabal al Gharbi	North	Outpatients	25% (IgG)	[63]
Egypt	SFNV	1950’s	NA	NA	NA	NA	Febrile patients	NA	[76]
		1952–1954	Qalyub	NA	Qalyubia	North	Humans	31% (Nt-Ab)	[38]
		1959–1961	Metropolitan Cairo	NA	Cairo Giza Qalyubia	North	*Ph. papatasi*	NA	[64]
		1960–1963	Giza	NA	Giza	North	Humans	43.8% (Nt-Ab)	[38]
			Alexandria	NA	Alexandria	North	Humans	21.6% (Nt-Ab)	
			Baltim-Borg Burulus	NA	Kafr El Sheikh	North	Humans	6.7% (Nt-Ab)	
			Luxor	NA	Luxor	South	Humans	56.3% (Nt-Ab)	
			Siwa	NA	Matruh	North	Humans	3.9% (Nt-Ab)	
			Sidi Barrani	NA	Matruh	North	Humans	9.5% (Nt-Ab)	
			Bahig	NA	Alexandria	North	Humans	25% (Nt-Ab)	
			El Daba	NA	Matruh	North	Humans	13.9% (Nt-Ab)	
			Aswan	NA	Aswan	South	Humans	47.3% (Nt-Ab)	
		1975	Cairo	NA	Cairo	North	Humans	6.1% (Nt-Ab)	
		1984	Imbaba	NA	Giza	North	Hospitalized patients	2/55 (IgG)	[77]
		1991	Bilbeis	NA	Sharqiya	North	Humans	2% (IgG)	[78]
	SFSV	1952–1954	Qalyub	NA	Qalyubia	North	Humans	22.6% (Nt-Ab)	[38]
		1959–1961	Metropolitan Cairo	NA	Cairo Giza Qalyubia	North	*Ph. papatasi*	NA	[64]
		1960–1963	Giza	NA	Giza	North	Humans	43.8% (Nt-Ab)	[38]
			Alexandria	NA	Alexandria	North	Humans	27% (Nt-Ab)	
			Baltim-Borg Burulus	NA	Kafr El Sheikh	North	Humans	8.9% (Nt-Ab)	
			Luxor	NA	Luxor	South	Humans	59.4% (Nt-Ab)	
			Siwa	NA	Matruh	North	Humans	2% (Nt-Ab)	
			Sidi Barrani	NA	Matruh	North	Humans	4.8%(Nt-Ab)	
			Bahig	NA	Alexandria	North	Humans	6.3% (Nt-Ab)	
			El Daba	NA	Matruh	North	Humans	8.3% (Nt-Ab)	
			Aswan	NA	Aswan	South	Humans	43.6% (Nt-Ab)	
		1969	NA	NA	NA	NA	University students	23% (IgG)	[79]
		1975	Cairo	NA	Cairo	North	Humans	5.0% (Nt-Ab)	[38]
		1984	Imbaba	NA	Giza	North	Hospitalized patients	1/55 (IgG)	[77]
		1989	Bilbeis	NA	Sharqiya	North	Schoolchildren	9% (IgG)	[80]
		1991	Bilbeis	NA	Sharqiya	North	Humans	4% (IgG)	[78]

*: TOSV and antigenically related viruses; NA: not available; E: east; N: north; W: west; m: meter; Ref.: reference.

## 8. Conclusions

Sandfly-borne phleboviruses were encountered frequently and widely over North African region. Synthesis findings counted eight evidenced phleboviruses and noted their geographical distribution. Their circulation became permanent, apparently due to virus maintenance by successive passages through vectors and vertebrate hosts. Their occurrence was indeed stated in Libya and Egypt, although respective knowledge was weak and evidently needs to be reinforced and updated. Noticeably, all studies conducted in Morocco were mostly interested in TOSV in sandflies, demonstrating its presence all over the country. Hence, no current data were available for TOSV in humans and animals, nor for any other sandfly-borne phlebovirus. Conversely, various sandfly-borne phleboviruses were found to cocirculate in both Algeria and Tunisia, mainly SFSV, TOSV, and PUNV. Phlebovirus investigations there concerned several vertebrate host species besides the vectors and showed that they were highly concentrated in the north. In addition, they have been recognized as potential etiological agents of neurological disease in humans. Additionally, TOSV activity was repeatedly reported and symptomatically confirmed in Algeria and Tunisia. Therefore, they should be considered for diagnosis in individuals with compatible clinical presentation. Further studies are required, not only to confirm the sporadic detection of less-prevalent phleboviruses, i.e., CYPV, SFNV, and MVV, but to fully elucidate the epidemiological situation in Morocco, Libya, and Egypt as well.

## Figures and Tables

**Figure 1 insects-15-00846-f001:**
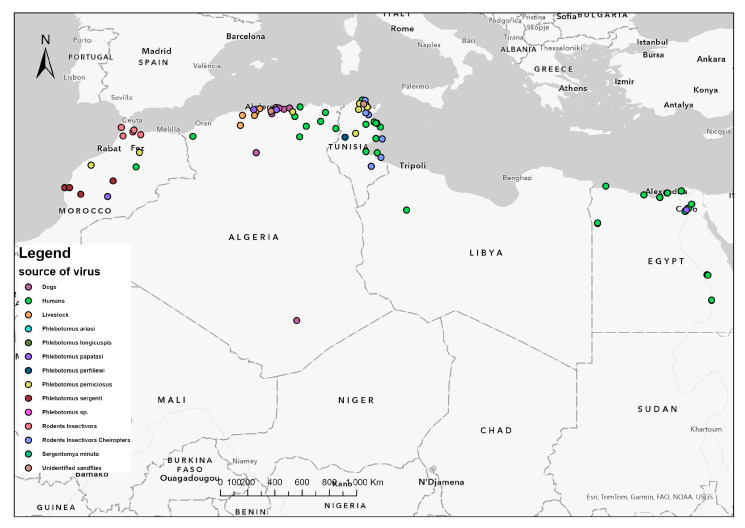
Distribution of detected and isolated sandfly-borne phleboviruses in vectors and vertebrate hosts from North Africa. A map of five North African countries showing geographical location of sandflies and vertebrates infected by sandfly-borne phleboviruses. Dots denote infected areas. Colors indicate species of sandflies, human, and non-human vertebrate hosts from the region.

**Figure 2 insects-15-00846-f002:**
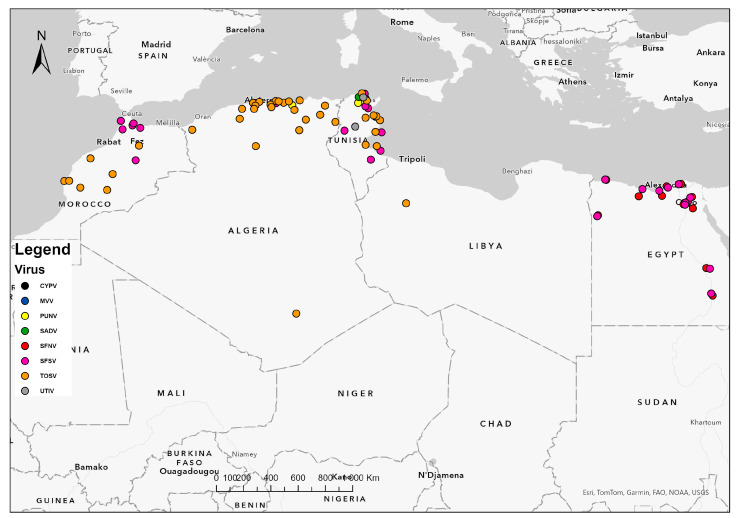
Geographical distribution of detected and isolated sandfly-borne phleboviruses in North Africa. A map of five North African countries showing geographical distribution of circulating sandfly-borne phleboviruses. Dots denote infected areas. Colors indicate species variety of detected and isolated sandfly-borne phleboviruses in the region. Legend: CYPV: Cyprus virus, MVV: Medjerda Valley virus, PUNV: Punique virus, SADV: Saddaguia virus, SFNV: Sandfly Fever Naples virus, SFSV: Sandfly Fever Sicilian Virus, TOSV: Toscana Virus, UTIV: Utique virus.

## Data Availability

No new data were created or analyzed in this study.

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
