# Peer review of "Epidemiology of Sandfly-Borne Phleboviruses in North Africa: An Overview"

_insects, 2024, doi:10.3390/insects15110846_

Round 1
Reviewer 1 Report
Comments and Suggestions for Authors
As a descriptive review it contains an exhaustive search of information, updating and integrating it.
In the figures, in order to integrate the information and contribute to risk mapping studies. a) Include in graph maps of known distribution of vectors and/or biomes, in figure 2, As suggested on page 16 ‘Their geographic distribution was quite similar and appeared to parallel that of the vectors occurring in humid zones (Figure.1), i.e. northern areas characterized by Mediterranean climate, or highly irrigated arid areas’. However, it is not shown in either figure. b) The dots represent sites, it could be more informative and integrative, to understand hot spots, if study areas are defined. c) Also include pseudo-absence areas (virus searched and not detected)
Discussion, in order to contribute to the analytical and not only descriptive discussion a) There is repetition of information in results, table and discussion, reduce redundancy. b) Discuss in a comprehensive manner, possible prevalence biases due to differences in protocol and sampling effort of the works cited, for example in vector protocol, proximity to food sources that can amplify or are refractory to the virus; in effort, representativeness of the numerator (prevalence in sample vs. prevalence in population-time). c) Add a commentary on diagnostic methodologies used and whether any have specificity risks, especially in the face of new viruses being described. In that sense, condense information from other regions, outside the scope of the review. d) In conclusion, the ‘Further studies’ statement would be enriched by porposing more specific research priorities and protocols to determine the risk factors and levels of risk to humans.
Formal: a) The first time viruses are cited in the body of the text, they should be written with the full name not just the acronym (page 2), even if they have been previously written in the bastract. b) The same is necessary for the first time a genus of Phlebotominae is named with the specific binomial name (page 3). c) On page 13 P. perniciosus, the species is written with a capital letter.
Author Response
RESPONSE
Manuscript ID: insects-3255566
Type of manuscript: Review
Title: Epidemiology of sandfly-borne phleboviruses in North Africa: an overview
Authors: Sabrina SELLALI, Ismail LAFRI, Rafik GARNI, Hemza Manseur, Mohamed BESBACI, Mohamed LAFRI, IDIR BITAM *
Received: 27 Sep 2024
Response to reviewers’ comments:
We thank the reviewers for their relevant remarks aiming to eliminate confusion and improve the clarity and explicitness of our paper, and we totally agree with them.
Following the reviewer recommendation, the correct abbreviations (Ph., Le., Se., and spp.) have been replaced everywhere in the text.
As asked by the reviewer, the italic form is now omitted from the word “Diptera” in line: 52.
As inquired by the reviewer, we dedicated the paragraphs below to improve and deepen the discussion (lines 502-533):
“The discrepant seroprevalence would be due to the vector distribution among the highly changing sampling areas, which is tightly related to geographical area, and environmental factors [31, 32, 45]. Sandfly borne phleboviruses occurred in the arid and humid bioclimatic stages, but restricted mainly to the north. It is strongly suggested that phleboviruses are heavily affecting highly populated regions, like the northern and the coastal, whose sandfly-exposed inhabitants are at greater risk of infection [31, 34]. Their distribution is congruent with entomological data. Sandflies are abundant in peri-urban and rural environments, and are often found in close proximity to human and domestic animal habitats, which results in an overlap [35, 45].
However, not all persons in a community have equal exposure to sandflies; some people apparently escape infection. Sandflies often show a-marked atmosphere preference. This may indicate that some individuals within a community are at greater risk of virus infection than others [38]. In the Maghreb region, a total of 32 species of the genera Phlebotomus and Sergentomyia were reported (15 from Libya, 18 from Tunisia, 23 from Mo-rocco, and 24 from Algeria). They feed on a wide range of vertebrate hosts, and their trophic preferences vary depending on the sandfly species. Multiple feeding patterns were detected, with various combinations of humans and vertebrate hosts [64]. As example, Ph. perniciosus is more anthropophilic compared to Ph. longicuspis, which is more zoophilic, leading to a difference in human versus dog biting rates and subsequently to a difference in the infection status with phleboviruses [32].
More significantly, sandfly fauna density and diversity vary by bioclimatic zones [54]. The number of sandfly vectors, namely: Ph. sergenti, Ph. longicuspis and Ph. perniciosus showed a positive correlation with altitude. In fact, the biodiversity index and richness were higher in highland areas (912 m above sea level) than in the plain area. Along with the abiotic factor of altitude, the distribution, biodiversity and richness of sandflies seemed primarily influenced by the bioclimatic factors such as the effect of soil texture on their habitats [11].
In addition, latest findings showed that environmental changes mainly due to the intensive development of irrigation systems for agriculture implemented in arid bio-geographical areas yielded to sustained populations of sandflies of the subgenus Larroussius mainly Ph. perfiliewi, Ph. perniciosus, and Ph. Longicuspis, which may have led to the emergence of phleboviruses in new areas [25, 60].”
Table1 has been completed by site factors, including altitude, latitude, longitude, and bioclimatic stage of sampling areas (when they are available in literature), as demanded.
As recommended, the abbreviations regarding the legends of figure1 are defined, in lines 493-495.
Reviewer 2 Report
Comments and Suggestions for Authors
Dear authors, review the comments annexed to the pdf MS.

Author Response

(The authors gave the same response as above.)

Round 2
Reviewer 1 Report
Comments and Suggestions for Authors
No further comments